# Mortality from Nonmelanoma Skin Cancer in Australia from 1971 to 2021

**DOI:** 10.3390/cancers16050867

**Published:** 2024-02-21

**Authors:** D. Czarnecki

**Affiliations:** Doctors’Care Clinic, 157 Scoresby Rd, Boronia, VIC 3155, Australia; dbczarnecki@gmail.com; Tel.: +61-97292356

**Keywords:** nonmelanoma skin cancer, mortality, population change

## Abstract

**Simple Summary:**

The study examined the number of deaths from nonmelanoma skin cancer (NMSC) in Australia for the fifty years from 1971 to 2021. Australia has the highest reported incidences of NMSC in the world. Deaths from NMSC have been recorded since 1971 and have increased more than five-fold in the 50 years to 2021. There is no sign of a reduction in the increasing incidence of deaths from NMSC. Most deaths from NMSC are due to cutaneous squamous cell carcinoma (SCC). It is estimated that 1 in 260 cutaneous SCCs will metastasize and cause death.

**Abstract:**

The number of non-melanoma skin cancers (NMSC) removed from Australians is increasing every year. The number of deaths from NMSC is increasing but so is the population. However, the population has greatly changed with many dark-skinned people migrating to Australia. These people are at low risk for skin cancer even if they live all their lives in Australia. The susceptible population is the rest of the population. The death rate from NMSC for the entire population and susceptible populations since 1971 is examined in this article. Materials and methods: Data on the Australian population were obtained from the Australian Bureau of Statistics (ABS). Every five years a census is held in Australia and detailed information of the population is provided. The ABS also provided yearly data on the causes of death in Australia. Results: The total population increased from 12,755,638 in 1971 to 25,738,140 in 2021. However, the susceptible population increased by far less, from 12,493,780 to 19,773,783. The number of deaths from NMSC increased from 143 to 765. The crude death rate for the susceptible population increased from 1.1 per 100,000 to 3.9 per 100,000. The crude death rate in the susceptible population aged 65 or more increased from 9.4 to 18.2 per 100,000. Conclusion: Deaths from NMSC are increasing despite public health campaigns to prevent skin cancer. According to current trends, NMSC will cause more deaths than melanoma in Australia.

## 1. Introduction

Non-melanoma skin cancer (NMSC) is a major health problem in Australia. The incidence of NMSC is unknown because cases are not recorded by State Cancer Registries except in Tasmania, which is the least populated state [1]. However, the magnitude of the problem can be ascertained by examining data from Medicare, the Australian Health Insurance Agency. Medicare records data on all patients treated in private medical practices in Australia. In 2021, Medicare recorded that 890,000 histologically confirmed NMSCs were excised from Australians and the number removed is increasing by 2% a year [2]. The Medicare data record the number of cancers removed not the number of people with NMSC. The incidence cannot be calculated from Medicare data because some people had more than one NMSC removed [3].

Deaths from NMSC are uncommon and have been recorded in Australia since 1971. The cause of death is obtained from death certificates and an Australian study found that there was little misclassification of the cause of death when it was due to NMSC [4]. The death certificates do not record what type of NMSC caused the death but squamous cell carcinomas (SCC) cause most NMSC deaths [1,4]. Cutaneous SCCs metastasize relatively frequently. Australian studies have found that 4.5% to 5.8% of cutaneous SCCs metastasize [5,6]. Similar percentages of metastases have been found in other regions of the world [7,8,9]. Basal cell carcinomas (BCC) on the other hand rarely metastasize. In Australia, BCCs metastasize in approximately 1 in 35,000 cases [10].

The risk of developing NMSC is very low for certain races, even if they live in a tropical environment where ultraviolet light levels are high for much of the year. In tropical Australia, the incidence of NMSC in the white population is 300 times higher than in Chinese, Indians, and Malays living in tropical Singapore [11,12]. In temperate England, white people are 475 times as likely to develop a cutaneous SCC as their Indian fellow citizens, while Chinese citizens living in England are 3900 times less likely to develop cutaneous SCC than whites [13]. In Asian, Arab, and African populations NMSC is rare [14,15,16,17]. Studies on the incidence and mortality from NMSC have to take into account changes in the racial composition of the population over time, otherwise, misleading conclusions about trends will be obtained. As more people at low risk for skin cancer settle in a country the incidence of skin cancer will be lower if it is calculated for the entire population. Epidemiologists have to calculate the number of susceptible people in each age group and for each year in order to obtain accurate trends on incidence and mortality.

In this report, the trend in the incidence of mortality from NMSC in Australia is examined for the census years from 1971, when the data were first recorded, until the last census in 2021. Data are provided for the entire population and the susceptible population.

## 2. Materials and Methods

Data on the Australian population were obtained from the Australian Bureau of Statistics (ABS). A census is held every five years and people are obliged by law to answer the questions. The countries of birth of the population were recorded in all censuses but the ancestry of the population was not recorded until the 2001 census [18]. Each year the ABS provides data on all births and deaths in Australia for the calendar year. The country of birth of the parents of Australian-born children is recorded but not ancestry. Where skin cancer was the cause of each death, it is listed as either melanoma being the cause or nonmelanoma skin cancer. The type of NMSC is not stated [1]. However, the number of deaths caused by cutaneous SCCs can be estimated from a Western Australian study of NMSC deaths [4].

The people at low risk for NMSC are Australians who were born in Asia, the Pacific Islands, the Middle East, or sub-Saharan Africa (low-risk regions). Australian-born children of people who migrated from these low-risk regions also have a low risk for all types of skin cancer. Living in Australia does not increase the risk of developing skin cancer in these populations [19]. The susceptible population is the total population minus the low-risk population.

The incidence of deaths from NMSC was determined for the entire population, the susceptible population, and the population aged 65 years or more. This is because NMSC is much more common in elderly people. In temperate Australia the average age at which an SCC develops is 72 years [20]. This is similar to the median age to develop SCC in temperate New Zealand [21]. Even in tropical Australia SCC is not common in young people and the median age at which an SCC develops is 66 years [11]. The Age Standardized Rate (ASR) of death for the entire population is added for comparison. The ASR is based on the 2001 standard Australian population. Data on ASR death rates for those aged 65 years or more are not available.

## 3. Results

The entire Australian population increased by 12,982,502 between 1971 and 2021 (see Table 1). A very large percentage of the increase was due to immigration. From 1971 to 2021, the net overseas migration into Australia was nearly 6,200,000. China and India were the countries from where most immigrants came. There was a major increase in net migration in 2007. Between 1971 and 2006, the average annual net intake of immigrants was 92,000. Between 2007 and 2021, the annual net intake increased to 227,000 people.

The entire population increased by 90% from 12,755,638 to 25,738,140, but the susceptible population increased by 54%, from 12,493,780 to 19,240,997. By 2021, 16.6% (4,267,760 people) of the Australian population were born in low-risk regions. There were also 1,676,597 children born in Australia whose parents migrated from the low-risk regions of the world and these children also have a low risk for skin cancer.

In the population aged 65 years or more, there was a smaller difference between the total and susceptible populations (see Table 2). This is because most immigrants from low-risk regions are young. In the 2021 census, 40% of immigrants from regions with a low risk for skin cancer were under the age of 40 years and 52% were under the age of 50 years. Only 11.6% of immigrants from these regions were aged 65 years or more. In addition, the Australian-born children of immigrants from low-risk regions were overwhelmingly aged under 40 years [18]. In contrast, only 20.4% of European-born immigrants were under the age of 40 years, 31.7% were under the age of 50 years and 42.7% were aged 65 years or more.

The number of deaths due to NMSC increased by 434%, from 143 to 765. The increase in deaths was greater in women than in men. In 1971 there were 44 female deaths from NMSC compared to 251 in 2021, a 578% increase. For men, the deaths increased from 99 to 514, a 419% increase. Although the increase in deaths was higher in women it was not statistically significant. In those aged 65 years or more, the increase in NMSC was 530% in women and 634% in men. The difference was not statistically significant.

The crude death rate from NMSC for the entire population increased from 1.1 to 3.0 per 100,000 people since 1971. This was the same as the increase in the ASR of mortality. However, there was a greater increase in the crude death rate for the susceptible population, from 1.1 to 3.9 per 100,000 people. This was 30% higher than the ASR for the total population and emphasizes that adjustments have to be made to only include susceptible people when calculating the rates of skin cancer deaths; otherwise, the rates will be lower. The crude death rate was stable between 1996 and 2006 for unknown reasons then steadily increased and shows no signs of stabilizing.

The crude death rate from NMSC was higher in the elderly population. It increased from 9.3 to 16 per 100,000 people for the entire population aged 65 years or over, but from 9.4 to 18.2 per 100,000 for the susceptible population in this age group. The crude death rate would be 13% higher if adjustments were made to the population base to only include susceptible people. The crude death rate was stable between 1981 and 1991, again for unknown reasons, then the rate steadily increased.

## 4. Discussion

The number of deaths from NMSC increased by more than five-fold between 1971 and 2021, whereas the susceptible population increased by only 60%. The trend for increasing NMSC deaths shows no signs of slowing. The greatest increase in deaths was in elderly Australians. In those aged 65 years or more, the number of deaths increased seven-fold. This was almost double the rate of increase in the susceptible population.

There were few NMSC deaths in Australians under 65 years but the number increased by 56% over 50 years, from 44 to 69. This was higher than the increase in the susceptible population, which was 40%, indicating that the public health problem of NMSC is not improving and that it is not only the ageing of the population that is the cause of the increase in NMSC mortality rates. It is important to note that the total population under 65 years increased by 83%. If the death rate from NMSC was calculated for the total younger population, there would be a decrease and it would appear that public health campaigns are reducing NMSC deaths.

There were few deaths from NMSC in Australians aged under 40 years but the number has not decreased despite these Australians being exposed to public health campaigns for their entire lives. In 1971, there were three deaths in a susceptible population of 8,184,656, and in 2021 there were seven deaths in a susceptible population of 9,345,087. The crude rate of deaths from NMSC doubled in 50 years but the numbers are too small to draw conclusions about the effectiveness of public health campaigns in preventing NMSC.

The number of deaths from NMSC is increasing by an average of 11% a year while the crude death rate is increasing by an average of 7% a year. The trends are not decreasing and could become worse because of restricted medical services from 2020 to 2022. During these years there were several lockdowns in Australian cities because of coronavirus. People were confined to homes for up to 23 h a day. Access to medical services in private practice and in public hospitals was restricted, and private hospitals were closed. Many elderly Australians were too afraid to attend medical treatment. These restrictions might have caused a reduction in the number of skin cancers diagnosed and treated at an early stage during these three years. The number of deaths from NMSC might greatly increase from 2023, when restrictions were lifted, because the pandemic was declared to be over, and medical services returned to normal. It is as yet unknown if patients are presenting to doctors with thicker SCCs, ones that are more likely to metastasize.

The crude rate of NMSC for the susceptible population is a more accurate way of measuring trends for skin cancer than the ASR for the entire population. This is because race is a far more important risk factor for skin cancer than age [7,11,12]. Australian epidemiologists have not calculated the percentage of susceptible people in each age group for different periods since 1971. This means that there is no standard susceptible population to compare skin cancer rates. The ASRs for melanoma incidence, melanoma deaths, NMSC incidences, and NMSC deaths are calculated using the 2001 Australian Standard Population. The 2001 population is vastly different from the 2021 population and using it to calculate trends in incidence rates gives a misleading picture of what is happening in Australia with regard to skin cancer [22]. Authors have claimed that mortality from NMSC is stable in Australia, based on the ASR of death over time [23], whereas the crude death rate shows no stabilizing trend.

The racial composition of the population is rapidly changing because Australia has a high immigration intake. In 2022, more than 400,000 immigrants were admitted into the country, most of them from Asia [24]. They have a low risk for NMSC. The migrants from Europe will also have a lower risk for skin cancer because they have not been exposed to as much ultraviolet light during their lives. The age of migration to Australia has been shown to be a risk factor for skin cancer [25]. People who migrate from Europe after childhood have a lower risk for skin cancer than the native-born population or people who migrate early in childhood.

Migration has already influenced the reports on the incidence of melanoma in Australia. Cases of invasive melanoma have been registered in Australia since 1982. In recent years claims have been made that the incidence of invasive melanoma is decreasing in young Australians. These authors did not take into account the great change in the racial composition of the population with a large percentage of the population now being at low risk for melanoma [22,26]. The incidence of invasive melanoma is decreasing for the entire population but if adjustments are made to the population base, the incidence of melanoma is increasing in all susceptible Australians irrespective of age [22].

Squamous cell carcinoma is the main cause of death from NMSC. An Australian study found that 70% of deaths from NMSC were due to cutaneous SCCs and 14% due to Merkel cell carcinoma [4]. Based on this study’s findings, it is estimated that in 2021, cutaneous SCCs caused 536 deaths in Australia. In the three years 2019 to 2021, cutaneous SCCs caused an estimated 1516 deaths. Medicare recorded 2,505,096 NMSCs being removed from Australians. There are approximately five BCCs removed for every SCC removed in Australia [20]. Based on these data, between 2019 and 2021 there were approximately 420,000 cutaneous SCCs removed and approximately 1 in 275 SCCs caused a death within three years [2,4]. This was lower than in England where a study found that about 1 in 150 cutaneous SCCs cause death within three years [13]. The reasons for the differences are unknown but Australians are very conscious about skin cancer, because of public health campaigns, and might seek medical treatment earlier than the English. Excising a cutaneous SCC in its early stages means that it is less likely to metastasize.

Australian studies have found that cutaneous SCCs metastasize in about 5% of cases but the number of people dying due to SCC is far less. This is probably because SCCs mainly develop in elderly patients who have other diseases and they are more likely to die of another disease than skin cancer. In temperate Australia, the average age to develop an SCC is 72 years, while the average age of death from cutaneous SCC is 74 years for men and 81 years for women [3,27].

However, the incidence of cutaneous SCC is very high and is increasing. Studies based on regional Australian populations have found that the incidence of cutaneous SCC is much higher than originally estimated in a national survey of self-reported skin cancer. The 2002 survey found that the incidence of SCC was 387 per 100,000 people [28]. In Tasmania, the region furthest away from the equator and the only state that registers all cases of NMSC, the incidence of cutaneous SCC was 514 per 100,000 in 2018 [29]. A study in a city in the coastal region of New South Wales, which is in the temperate zone of Australia, found that the incidence of histologically confirmed cutaneous SCC was 865 per 100,00 people between 2016 and 2018 [30]. In tropical Australia, the incidence of histologically confirmed SCC was the highest in the world at 1332 per 100,000 men [11]. The Tasmanian Cancer Registry data have found that the incidence is increasing by 6% a year.

The increasing incidence of NMSC is a worldwide trend. The increase has been reported from countries with different climates such as New Zealand and Russia [31,32] and different races such as Japan, Hong Kong, and South Africa [14,15,16]. However, the incidences in Asian countries and in black South Africans are still far lower than in white populations. If the trend of increasing incidence of NMSC continues, cutaneous SCCs will eventually be the cause of more deaths than cutaneous melanoma because it will be far more common and more prevalent.

The two weaknesses of this study are that the cause of death was determined by death certificates and that the country of birth had to be used as a surrogate for race. The data from death certificates have been determined to have few mistakes and give an accurate picture of the cause of death in Australia [4]. The country of birth has to be used as a surrogate for race because the census data do not include ancestry. The census data show that there has been a great change in the Australian population and this change has influenced data on skin cancer incidence in Australia over the last 50 years.

## 5. Conclusions

These data show that there is no decrease in the trend for increasing incidence of NMSC in Australia and the number of deaths caused by NMSC has greatly increased since 1971. Currently, it is estimated that 70% of susceptible Australians will develop a NMSC during their lifetimes [28]. If this trend continues, NMSC will cause more deaths than cutaneous melanoma because NMSC is far more common than melanoma. Australian public health campaigns aimed at reducing the incidence of all types of skin cancer have been running since 1980. They have not worked to date and need to be changed. However, epidemiologists have not realized that skin cancer is increasing in susceptible Australians and no studies have been undertaken to find out why the health campaigns have failed.

## Figures and Tables

**Table 1 cancers-16-00867-t001:** The Death Rates of from NMSC—Total Australian Population.

Census Year	Total Population	Deaths NMSC	ASR *	Crude Rate **	Susceptible Population	Crude Rate ***
1971	12,755,638	143	1.1	1.1	12,493,780	1.1
1976	13,514,947	153	1.1	1.0	13,176,469	1.2
1981	14,576,330	192	1.3	1.3	13,999,548	1.4
1986	15,602,155	206	1.3	1.3	14,810,285	1.4
1991	16,850,540	266	1.5	1.6	15,614,435	1.7
1996	18,224,970	359	2.0	2.0	16,240,220	2.2
2001	19,274,700	387	2.0	2.0	16,775,838	2.3
2006	20,450,970	410	2.0	2.0	17,874,068	2.3
2011	22,340,020	532	2.4	2.4	18,791,824	2.8
2016	24,127,160	681	2.8	2.8	19,240,997	3.5
2021	25,738,140	765	3.0	3.0	19,773,783	3.9

* Age Standardized Rate (2001 Australian population); ** Crude rate per 100,000 total population; *** Crude rate per 100,000 susceptible population.

**Table 2 cancers-16-00867-t002:** The crude rate of deaths from NMSC—People aged 65 or more.

Census Year	Total Population	Deaths NMSC	Crude Rate *	Susceptible Population	Crude Rate **
1971	1,064,995	99	9.3	1,047,651	9.4
1976	1,208,980	109	9.0	1,193,362	9.1
1981	1,429,400	135	9.4	1,404,305	9.6
1986	1,646,719	155	9.4	1,619,202	9.6
1991	1,906,741	179	9.4	1,878,693	9.5
1996	2,192,224	304	13.9	2,093,056	14.5
2001	2,419,070	335	13.8	2,288,829	14.6
2006	2,664,060	342	12.8	2,496,584	13.7
2011	3,087,910	456	14.8	2,856,356	15.9
2016	3,672,250	586	15.9	3,332,345	17.6
2021	4,328,860	696	16.0	3,816,860	18.2

* Crude rate per 100,000 total population, ** Crude rate per 100,000 susceptible population.

## Data Availability

Data sharing is not applicable to this article.

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
