# Peer review of "Mortality from Nonmelanoma Skin Cancer in Australia from 1971 to 2021"

_cancers, 2024, doi:10.3390/cancers16050867_

Round 1

Reviewer 1 Report

Comments and Suggestions for Authors

In future investigation, in order to avoid the two weakness of the study, some new source of data should be used.  

Author Response

no response

Reviewer 2 Report

Comments and Suggestions for Authors

1. Introduction

The introduction provides a comprehensive overview of NMSC's significance in Australia, its epidemiology, and its associated challenges. Minor grammatical adjustments and refinements for clarity can enhance the presentation of the information.

2. Materials and Methods

The ” Materials and Methods ” section is quite clear and provides a structured explanation of the materials and methods used in the study. The information is organized logically, detailing the sources of data, the criteria for defining low-risk populations, and the specific age groups considered.

3. Results

The data presented seems specific and detailed, suggesting authenticity. The text is generally well-structured but can benefit from grammatical adjustments and clarity improvements.

4. Discussion

The content is clear. Consider refining for flow: "...the Census data indicate significant demographic shifts influencing skin cancer incidence trends." (lines 226-229)

Overall, this section comprehensively analyzes the trends and factors influencing NMSC incidence and mortality in Australia. Some minor grammatical adjustments and refinements for clarity can enhance the presentation of this section.

5. Conclusion

The content is clear and to the point. A slight rephrasing might improve clarity: "These findings suggest that the existing Australian public health campaigns aimed at reducing skin cancer incidence, initiated in 1980, have been ineffective thus far and require revision."

Congratulations to the author for his work and contribution to the field!

Comments on the Quality of English Language

1. Introduction

The introduction provides a comprehensive overview of NMSC's significance in Australia, its epidemiology, and its associated challenges. Minor grammatical adjustments and refinements for clarity can enhance the presentation of the information.

2. Materials and Methods

The ” Materials and Methods ” section is quite clear and provides a structured explanation of the materials and methods used in the study. The information is organized logically, detailing the sources of data, the criteria for defining low-risk populations, and the specific age groups considered.

3. Results

The data presented seems specific and detailed, suggesting authenticity. The text is generally well-structured but can benefit from grammatical adjustments and clarity improvements.

4. Discussion

The content is clear. Consider refining for flow: "...the Census data indicate significant demographic shifts influencing skin cancer incidence trends." (lines 226-229)

Overall, this section comprehensively analyzes the trends and factors influencing NMSC incidence and mortality in Australia. Some minor grammatical adjustments and refinements for clarity can enhance the presentation of this section.

5. Conclusion

The content is clear and to the point. A slight rephrasing might improve clarity: "These findings suggest that the existing Australian public health campaigns aimed at reducing skin cancer incidence, initiated in 1980, have been ineffective thus far and require revision."

Congratulations to the author for his work and contribution to the field!

Author Response

minor adjustments have been made

Reviewer 3 Report

Comments and Suggestions for Authors

This study evaluate the mortality from nonmelanoma skin cancer (NMSC)  in Australia from 1971 to 2021 and had signficant findings. Overall, the study is well-designed and the manuscript is well-written. Thus, I have only three minor suggestions.

1. Beccause the findings were limited, a brief report may be appropriate for this stubmisison.

2.  I did not know why the author showed the data every five year rather every year. In addition, please add the figures to show the annunal death rate.

3. I wonder whether the death rate among susceptible population was calculated  based on the number of deaths of only susceptible patients or the overall population.  Please clarify this issue for both susceptible and aged populations.

Author Response

The journal editors wanted a 3000 word article.       The data are for Census years. A census is held every 5 years and the questions must be answered or a hefty fine is imposed. The Australian Bureau of Statistics (ABS) does random surveys of the population after the census and is confident that the data are accurate. In between the censuses the Bureau of Statistics gives estimates of the population. The data are for the census years because these data are accurate. The death rates were based all deaths because the race of the dead person is not given on the certificate. The death rates are based on all deaths fro NMSC because the race of the deceased is not given on the certificates.

Reviewer 4 Report

Comments and Suggestions for Authors

The short manuscript reports findings from a trend analysis of mortality data from non-melanoma skin cancer (NMSC) over a fifty year period from 1971 till 2021. The information on NMSC mortality is based on official data from death certificates and is related to census data of the Australian Bureau of Statistics to calculate (crude) mortality rates.

It is not the first time that the author addresses this topic to get the message across that the impact of immigration has to be taken into account when interpreting mortality trends in Australia. He has published the same message several times and now echoes his earlier papers and letters using updated data until 2021. I agree on his argument that raw numbers of NMSC deaths should be related to what he calls the susceptible population rather than to the total population to better reflect mortality changes in such trend analyses. However, I do see a methodological problem in his calculations that ignore any change in the age distribution of the susceptible (and total) population over time. The NMSC mortality rates calculated are not age-standardized which is a clear methodological drawback and may jeopardize comparisons over time. This critique applies even to the age-restricted analysis of the older subpopulation (Table 2), as the subpopulation aged 65 years and above will also have changed its age distribution over the last 50 years due to the increase in life expectancy.

In a major revision the author should replace the crude NMSC rates in Table 1 and Table 2 by age-adjusted NMSC rates (or give both rates). The decision of which age distribution is chosen for the standardization will have an impact on the numbers, but the evaluation of the trend in NMSC rates does not depend critically on the chosen standard. If the trend in age-standardized NMSC mortality is consistent with what is seen in crude NMSC mortality rates over time, then this would strengthen the author's conclusion.

The manuscript acknowledges its limitations with respect to its inability to differentiate between different forms of NMSC (SCC, BCC, Merkel cell carcinoma etc.). However, it nevertheless tries to make quantitative statements about the number of SCC deaths in the Discussion section (line 186ff) derived from a simple projection based on data from a single study by Girschik et al.. Girschik et al's study comprised only 300 cases that were coded as NMSC in Australian death certificates and were subsequently reevaluated. Projections based on such a small study have a large margin of uncertainty which may explain discrepancies to the recent data from an English study by Venables et al.. The authors should make it clearer that he uses simple ad hoc projections and avoid authorative statements like "In the three years 2019 to 2021, cutaneous SCCs caused 1516 189 deaths" (line 189/190) when presenting the results of these projections.

In the Results section sex-specific results are mentioned only in the text (line 106-111). A more detailed presentation of sex-specific (age-adjusted) mortality data in (a) separate table(s) would be a welcome addition.

Overall, I liked the manuscript that has a good point in refering to the immigration problem when performing trend analyses. However, the effects of immigration and changes in the age distribution have to be disentangled to better understand the true dynamics of NMSC mortality over time.

Author Response

I keep writing about trends in skin cancer incidence and mortality in Australia because Australian epidemiologists, and Anti Cancer Council executives, are still claiming that the trends show that the incidence of skin cancer is decreasing in young Australians and that this is due to public health campaigns. They fail to mention the population change that has occurred. Currently about one quarter of Australians have a very low risk of developing skin cancer because they have dark skin - the low risk group. Skin colour is the major risk factor (or protective factor) for skin cancer. It is far more important than age. In Australia age standardized rates for skin cancer are based on the 2001 population. This is nothing like the 2021 population. Epidemiologists have not calculated the age/race standard population for any year. This would be the way to go to calculated trends for skin cancer.

I have included the age standardized rates for NMSC mortality in Table 1. The data are not available for the age group 65 or more. The text has been changed to acknowledge this.

The article now states that it is estimated that 1 in 275 cutaneous SCCs will kill.

A table showing sex specific mortality does not add anything to the article, in my opinion.

Reviewer 5 Report

Comments and Suggestions for Authors

The article is very interesting for readers.

Data related to the possible association at the same person between NMSC and Melanoma,between BCC and SCC or Merkel Cell Carcinoma are of interest if available.

Data related to sunscreen users versus non users in Australia are of interest,also about the afected area of the skin by NMSC,or smokers versus non smokers.

Data from Australia regarding some popular drugs like hidrochlorothiazide or statins related to NMSC or Melanoma,comparative with other populations like: 

Lin BM, Li WQ, Cho E, Curhan GC, Qureshi AA. Statin use and risk of skin cancer. J Am Acad Dermatol. 2018 Apr;78(4):682-693.

doi: 10.1016/j.jaad.2017.11.050

POTTEGARD A, HALLAS J, OLESEN M, et al. Hydrochlorothiazide use is strongly associated with risk of lip cancer. J Intern Med. June 2017. doi:10.1111/joim.12629

doi: 10.1111/jdv.15084

Those informations can be used and refered.

Author Response

Deaths from skin cancer are classified as from melanoma or non-melanoma skin cancers (NMSC).  All NMSC are grouped together but most deaths are caused by cutaneous squamous cell carcinoma. The Western Australian study found that 70% of NMSC deaths were due to squamous cell carcinomas and 14% due to Merkel cell carcinoma.

Data on sunscreen use by the deceased are not available as is the case with data on smoking and what drugs were taken.

Reviewer 6 Report

Comments and Suggestions for Authors

 Dear Author,

The manuscript is overall well written but it brings news strictly for your country.

However, I consider that an exhaustive review like this should be accompanied with some clinical spects of the disease.

Minor comments are as listed:

ü  table 2 is misspelled in the text

ü   you did not use square brackets when introducing references in the text

ü  all references are written without following the intructions

I recommend accepting after a minor revision.

Author Response

The article is about trends in death rates over 50 years. Clinical  aspects of skin cancer can be written in another article.

The tables had Roman numerals, they now have Arabic numerals. The references now have square brackets.

Reviewer 7 Report

Comments and Suggestions for Authors

"The paper by Czarnecki, entitled “Mortality from Nonmelanoma Skin Cancer in Australia from 1971 to 2021,” is an original article that discusses the epidemiology of non-melanoma skin cancer (NMSC) in Australia. The author's main conclusion is that deaths from NMSC are increasing despite public health campaigns to prevent skin cancer and that NMSC is projected to cause more deaths than melanoma in Australia, if the current trends continue. However, I have some concerns about the paper. 

Firstly, the author only provides crude rates of death from NMSC among the total population and susceptible population. I am uncertain if this data is sufficient for publication as an original article, and I believe it should be left up to the Editor to decide. 

Secondly, I would like the abstract to briefly explain what the author means by “susceptible populations.” 

Lastly, I think it would be interesting to discuss why the death rate from NMSC is increasing despite the public health campaigns."

Author Response

The age standardized death rates (ASDR) cannot be used because the population is rapidly changing. In 2023, Australia admitted 500,000 immigrants (for a population of 24 million. The Australian population of 2001 is nothing like the population of 2011, which is nothing like the population of 2016 etc. Skin cancer is overwhelmingly a disease of white people. If you want to use ASDR, you have to calculate the percentage of white people in each age group in each year. This had not been done. The ASDR is calculated for the entire population and because each year there are more an more people at low risk for skin cancer the data show a decreasing trend. The crude rate gives a more accurate picture of what is going on.

The death rate is increasing because the public health campaigns have not worked. the incidence of melanoma is increasing even in young susceptible Australians who have been raised with the public health campaigns. The references are in the bibliography

Round 2

Reviewer 4 Report

Comments and Suggestions for Authors

The change of NMSC mortality rates in a population over time can have different reasons. Everything else being constant a change of the age distribution over time towards more older people being part of the population will increase the NMSC mortality. Everything else being constant a change of the skin type distribution over time towards a higher proportion of people with dark skin will conversely decrease NMSC mortality. When the effect of other factors like public health campaigns on the development of NMSC mortality is of interest, then both, the change in the age distribution and the change in skin type distribution over time, should be adjusted for. Standardization is the typical (and easy-to-use) methodological tool to accomplish this.

The original manuscript reported in Table 1 only crude NMSC mortality rates over time in the total population and the susceptible subpopulation (excluding people with dark skin from the total population). The comparison of the temporal development in both populations was then the basis for concluding that the levelling-off of NMSC mortality in the total population (which ignores the effect of changes in the skin type distribution) does not tell the whole story as NMSC mortality is still increasing in the susceptible population. My criticism that a potential change in age distributions over time is not taken into account in this analysis was addressed in the revised version by adding age-standardized NMSC mortality figures (only) for the total population in Table 1. This expansion does, however, not address my criticism in a meaningful way. Age-standardized NMSC mortality rates in both populations (total and susceptible part) are required to be able to evaluate temporal changes validly. Otherwise, apples are compared with pears. Therefore, the author should calculate age-standardized NMSC mortality rates for the susceptible part of the population based on offical data and include this information in Table 1.

Author Response

I have replied to the reviewer about age standardized incidence rates and mortality rates. When it comes to skin cancer, race is the most important risk factor, not age. Epidemiologists will have to calculate the percentage of people with dark skin (low risk for skin cancer) in each age group and for each year. Yes, each year because the Australian population is rapidly changing. In 2023, 540,000 migrants arrived, which resulted in a 2% increase in the population. The 2023 population is different from the 2022 population. The 2021 population is vastly different from the 2001 population, which is the population used to calculate age standardized rates. 

    The data are set out in table 1. In 1971, 98% of the population was susceptible to skin cancer. In 2001, 87% of the population was susceptible to skin cancer. In 2021, 77% of the population was susceptible to skin cancer. In the first 30 years, the susceptible population fell by 11%. In the next 20 years it fell by 10%. That shows you how rapid are the changes in the population.

    Epidemiologists have not taken these population changes into consideration. No one has calculated an age-race susceptible population for any year. There is no age standardized population for susceptible Australians. The age standardized rates are for the entire population. Paragraph 5 in the discussion discusses this topic. I have changed the colour of the text to red.

   Table 2 shows how the NMSC mortality has risen far faster than the population aged 65 years or more. It is not the ageing of the population that is causing the increase. Australians have been exposed to public health campaigns since 1981 and if they were of benefit, NMSC should be stable or declining. They are not because susceptible Australians are not protecting themselves from too much sun exposure.

Reviewer 7 Report

Comments and Suggestions for Authors

Thank you for the revised version of the manuscript. I have no additional comment.

Author Response

Thank you.

Round 3

Reviewer 4 Report

Comments and Suggestions for Authors

Thank you for clarifying that you are not able to standardize NMSC rates in the susceptible population for age as you do not have access to the data necessary for standardization. I am not familiar with Australian census data and thus cannot evaluate whether these data do not exist or whether they are not accessible for you.

In essence, the article demonstrates that crude NMSC rates in the susceptible population in Australien show a different time trend than crude and age-standardized NMSC rates in the total Australian population. The methodologic problem that a time trend in crude NMSC rates may be influenced by temporal changes in the age distribution (which definitely have taken place in Australia over the last decades) could not be addressed and may be part of the explanation for the difference in time trends. This does not invalidate the analysis, but it makes the conclusion weaker (surprisingly, this not even mentioned in the paragraph devoted to the "weaknesses of the study" at the end of the Discussion section). To my opinion, it is inconsistent to criticise others with the methodological argument that when looking at temporal changes in NMSC rates, a restriction to the susceptible population is necessary to assess the actual temporal changes, while at the same time ignoring the problem of the changing age distribution and focusing on crude NMSC rates.

Author Response

Dear reviewer,

Once again, race is the main risk factor for skin cancer, not age. Age distribution over time is meaningless. Age-race distribution over time is meaningful but this has not been done. This fact is in the manuscript.
     The only new change is in the conclusion section. I have added a sentence pointing out that Australian epidemiologists have not detected that skin cancer is getting worse in white Australians therefore have not studied why the health campaigns have failed.
     The article is attached.
     Yours sincerely,
     Dr. D. Czarnecki